


# Gradient flux measurements of sea-air DMS transfer during the Surface Ocean Aerosol Production (SOAP) experiment

Murray J. Smith[1], Carolyn F. Walker[1], Thomas G. Bell[2,3], Mike J. Harvey[1], Eric S. Saltzman[2], Cliff S. Law[1,4]

[1]National Institute of Water and Atmospheric Research (NIWA), Wellington, 6241, New Zealand
[2]Earth System Science, University of California, Irvine, California, USA
[3]Plymouth Marine Laboratory, Plymouth, PL1 3DH, UK
[4]Department of Chemistry, University of Otago, Dunedin, New Zealand

*Correspondence to*: Murray J. Smith (m.smith@niwa.co.nz)

**Abstract.** Direct measurements of marine DMS fluxes are sparse, particularly in the Southern Ocean. The Surface Ocean Aerosol Production (SOAP) voyage in February-March 2012 examined the distribution and flux of dimethylsulfide (DMS) in a biologically-active frontal system in the southwest Pacific Ocean. Three distinct phytoplankton blooms were studied

with oceanic DMS concentrations as high as 25 nmol L$^{-1}$. Measurements of DMS fluxes were made using two independent methods: the eddy covariance (EC) technique using API-CIMS chemical ionization mass spectrometry, and the gradient flux technique (GF) from an autonomous catamaran platform. Catamaran flux measurements are relatively unaffected by air flow distortion and are made close to the water surface where gas gradients are largest. Flux measurements were complemented by near-surface hydrographic measurements to elucidate physical factors influencing DMS emission. Individual DMS fluxes

derived by EC showed significant scatter and, at times, consistent departures from the COARE gas exchange parameterization. A direct comparison between the two flux methods was carried out to separate instrumental effects from environmental effects, and showed good agreement with a regression slope of 0.96 (r$^2$ = 0.89).  A period of abnormal downward atmospheric heat flux enhanced near-surface ocean stratification and reduced turbulent exchange, during which GF and EC transfer velocities showed good agreement but modelled COAREG values were significantly higher. The transfer

velocity derived from near surface ocean turbulence measurements on a spar buoy compared well with the COAREG model in general, but showed less variation. This first direct comparison between EC and GF fluxes of DMS provides confidence in compilation of flux estimates from both techniques, and also in the stable periods when the observations are not well-predicted by the COAREG model.

## 1 Introduction

The transfer of gases across the air-sea interface has a significant influence on global climate, and so is an important parameter in climate models. Of particular interest is dimethylsulfide (DMS), a biogenic gas originating from phytoplankton,



which is the primary source of reactive natural sulfur in the atmospheric marine boundary layer (Andreae and Crutzen, 1997). The potential for oxidation products of DMS to play a role in aerosol production and modification, and hence the global radiation budget, has stimulated a number of studies aimed at quantifying the surface ocean concentrations and estimations of global DMS fluxes (e.g. Lana et al., 2011).

While climatologies of aqueous DMS have been developed (Lana et al., 2011), direct measurements of DMS fluxes are rare, particularly over the remote Southern Ocean. Instead, DMS fluxes typically rely on gas transfer parameterizations, with the flux of a gas ($F$) between the ocean and atmosphere calculated from the transfer relationship with concentration gradient across the interface:

$\quad F = K (Cw - \alpha Ca)$ (1)

where $K$ is the gas transfer velocity, $Ca$ and $Cw$ are the DMS concentration in air and water respectively, and $\alpha$ is the dimensionless Ostwald solubility coefficient for DMS (Dacey et al., 1984; Saltzman et al., 1993), with a positive flux indicating sea-to-air emission. In theory, the concentration gradient should be measured across the viscous sublayer at the surface (~100 µm), whereas in practice it is normally measured on research vessels between the air at deck level and the

seawater intake several meters below the surface.

In contrast to the measurement of near-surface gas concentration gradients, measurement and parameterization of the transfer velocity, $K$, is more challenging and subject to greater uncertainty, particularly at high wind speeds (Wanninkhof et al., 2009). $K$ has traditionally been parameterized in terms of wind speed, the most readily available measurement, and largely

based on dual tracer experiments using $SF_6$ and $^3He$ (Wanninkhof, 1992; Nightingale et al., 2000; Ho et al., 2006), most commonly in the form of a quadratic relationship. However, there are issues arising from the temporal and spatial scales of these and other estimates of $K$. For example, dual tracer experiments generally integrate over days and 100 km scales, bomb carbon estimates are decadal and cover basin scale, and micrometeorological estimates cover periods of <30 minutes and scales of several km. Consequently, processes operating on different scales may have different influences on $K$. In addition,

there are other factors influencing gas transfer. The role of bubbles in enhancing the transfer of insoluble gases, such as $CO_2$, is widely recognized (e.g. Woolf, 1997; Wanninkhof and McGillis, 1999), but while corrections for the diffusion rates are used for different gases via the Schmidt number, the difference in solubility between the two tracers and $CO_2$ is not generally considered (Bell et al., 2017). When simple wind speed parameterizations are applied to a weakly soluble gas such as DMS, the bubble effect is very much reduced relative to $CO_2$, and DMS is better fitted as a linear, rather than a quadratic,

relationship with wind speed (Blomquist et al., 2006; Goddijn-Murphy et al., 2012). In addition, $K$ depends on the resistance in the water and the atmosphere; Bell et al. (2015) estimated the atmospheric contribution for DMS as approximately 7 % of the total, whereas for less soluble gases such as $CO_2$, the airside contribution will be lower.





It is convenient to parameterize $K$ primarily in terms of wind speed, but the primary driver of gas exchange is near-surface turbulence, whether under a small-eddy surface renewal model (Lamont and Scott, 1970) or a mixing-length model (Donelan and Soloviev, 2016). For environments in which turbulence is driven by factors other than wind, such as benthic boundary layer turbulence in shallow estuarine flows or raindrop impacts, unifying approaches based upon eddy dissipation rate as a measure of turbulence, have been considered (Zappa et al., 2007), but measurements in the open ocean are rare (Esters et al., 2017).

The NOAA-COARE algorithm was developed to provide a more physically-based parameterization for gas fluxes, by separating the turbulent viscous stress from the wave-form stress related to bubble production, and has shown promising results for both $CO_2$ and DMS (Fairall et al., 2011; Yang et al., 2011). However, current understanding of the mechanisms of air-sea gas exchange are imprecise, and other physical processes may come into play that are not currently captured within the COARE model. For example, Marandino et al. (2008), encountered anomalously large DMS fluxes during a coccolithophore bloom in the North Atlantic, which they attributed to the presence of near-surface gradients.

Eddy covariance (EC) is the most direct method of measuring gas fluxes (Blomquist et al., 2006; Marandino et al., 2007) since it does not make assumptions about the structure of the turbulent boundary layer. However, EC requires rapid, high-precision measurements of gas concentration, which are then correlated with the vertical component of turbulent wind speed corrected for platform motion. The $k_{DMS}$ calculated from eddy covariance DMS fluxes during the Surface Ocean Aerosol Production (SOAP) experiment (Bell et al., 2015) showed considerable variability, which is typical of these measurements of a turbulent process. An ongoing issue is the extent to which the variability is inherent in the measurement technique, or due to other factors, and also whether it can be reduced. Bell et al. (2015) showed that some of the scatter in $k_{DMS}$ arose from spatial inhomogeneity of the seawater DMS concentration, but significant variability still remained. Wind and flux measurements are also challenging on a vessel since the airflow around the ship's superstructure can be accelerated or decelerated (Yelland et al., 1998), with a dependence on wind direction. This can be a significant factor where wind speed is an explicit parameter, such as with gas transfer velocities and drag coefficients. Other issues such as motion correction and airflow, can also influence shipboard flux measurement (Landwehr et al., 2015). An additional complication is the generation of turbulence by the interaction of air flow with the vessel's structure, such as the leading edge of the hull and/or smaller support structures (Oost et al., 1994). The effect of this on the apparent gas flux is not well known nor whether advected atmospheric turbulence is affected by flow distortion. Computational fluid dynamics (CFD) has been used in a number of studies seeking to quantify the magnitude of flow acceleration (Yelland et al., 1998; Popinet et al., 2004; Moat et al., 2005).

An alternative direct flux measurement approach is the Gradient Flux (GF) technique, which is less direct than EC as it depends on an assumption of Monin Obukhov Similarity (MOS) theory. The first open ocean measurements of DMS fluxes



using the GF method were made by Putaud and Nguyen (1996) with 2 or 3 air intakes on a ship. In order to avoid issues of the DMS gradient being influenced by air flow distortion over the superstructure of the research vessel, a platform has been used in other experiments, primarily in estuarine or coastal environments. McGillis et al. (2004) used a catamaran during GaseEx-2001 with a motorized traveller on a mast to sample a range of heights to generate $CO_2$ profiles, combined with $CO_2$

sampling from a boom on the ship's bow. A catamaran attached to a boom alongside a vessel, with a sensor travelling up the mast has also been used successfully in estuaries to determine the gas exchange of $CO_2$ (Orton et al., 2010; Zappa et al., 2007; Zappa et al., 2003). Although this approach has advantages in using the same sensor at all heights, it introduces uncertainty due to the lack of simultaneous sampling in a turbulent atmosphere. Recently, Omori et al. (2017) reported DMS gradient fluxes from a tethered buoy closeby with intakes running back to the ship and different heights sampled sequentially

using proton transfer reaction-mass spectrometry (PTR-MS).

In order to validate the GF technique for oceanic applications, flux estimates derived from GF have been compared with other direct and indirect techniques, with a variety of results. For example, Putaud and Nguyen (1996) report a GF flux parameterization 1.6 – 1.9 times greater than that of Liss and Merlivat (1986). Zemmelink et al. (2002b) trialled GF in

conjunction with the Relaxed Eddy Accumulation (REA) technique from a dock with promising results, although without the complications of ship motion. Hintsa et al. (2004) subsequently used the stable platform RP *FLIP* in the northeast Pacific to measure DMS fluxes using both GF and REA techniques. They found DMS gradient fluxes were half that of REA fluxes with the largest discrepancy during stable to neutral conditions. Zemmelink et al. (2004) also used the GF technique to measure DMS fluxes during GasEx-2001 in the equatorial Pacific under predominantly light winds. Their comparison of gas

transfer rates with EC fluxes of $CO_2$ showed a substantial difference, with a high degree of scatter that could not be accounted for, and they concluded it was not possible to derive accurate gas parametrizations from in-situ measurements. To reconcile these discrepancies, a direct comparison between GF and EC is desirable.

The Surface Ocean Aerosol Production (SOAP) experiment (Law et al., 2017) examined the role of surface ocean

biogeochemistry in influencing marine boundary layer aerosol, with a strategy of targeting phytoplankton blooms east of New Zealand, that were potentially significant source regions for aerosol precursors. These blooms represented a significant source of DMS (Bell et al., 2015), and provided a valuable opportunity for validation of micrometeorological techniques and parameterizations. Beyond the inherent importance of measuring DMS fluxes, the SOAP campaign also sought to intercompare sampling platforms and techniques. The EC DMS flux measurements during SOAP followed the mean trends

modelled by COARE, but when viewed in detail there were periods of significant departure between measured and modelled $K$ (Bell et al., 2015). Although significant sub-surface DMS gradients were not observed during SOAP, some DMS flux anomalies relative to COARE were potentially attributed to sea-surface microlayer production (Walker et al., 2016).



This paper focuses on GF DMS fluxes measured from a small autonomous catamaran launched from the research vessel. The data provide the opportunity to compare the fluxes modelled by COARE with two independent measurement techniques (GF and EC) on different platforms. Turbulence and near-surface ocean temperature were also measured from a drifting spar-buoy to provide supporting information on near-surface structure and stratification. This paper examines whether observations from the two approaches are in agreement, and also assesses in detail the contribution of physical conditions on the waterside to flux deviations from COARE. Finally, the practicality of using eddy dissipation rate as a proxy for gas transfer in the open ocean is considered.

## 2 Methods

The SOAP study was carried out on the RV *Tangaroa* during February and March 2012 in the biologically productive frontal waters of the Chatham Rise (44ºS, 174-181ºE), east of New Zealand, where nutrient-rich sub-Antarctic water meets warmer subtropical water (Law et al., 2017). The voyage strategy was to survey regional surface biogeochemical distributions during the night, and focus sampling during the day on the areas showing highest DMS concentrations, chlorophyll-*a* and $CO_2$ drawdown. A range of other biological and physical measurements were made from the vessel and platforms launched from the ship, to characterise the biogeochemical and hydrodynamic influences on trace gas emissions and aerosol production (Law et al., 2017).

### 2.1 Environmental measurements

The RV *Tangaroa* was equipped with an automatic weather station (AWS) mounted on top of the crowsnest above the bridge (Fig. 1a). While this position gave the clearest unobstructed airflow from all directions, it was still subject to some flow distortion as air was displaced and accelerated by the ship's superstructure. These effects were modelled in Popinet et al. (2004), and a correction for the airflow distortion for the RV *Tangaroa* as a function of azimuth was used in Smith et al. (2011). A data acquisition system (DAS), locked to GPS time (UTC), logged the main ships navigation parameters and a range of underway measurements such as sea surface temperature, chlorophyll fluorescence, and Ecotriplet β660 backscatter. Measurements are all time-referenced to day of year (DOY) in UTC. The COARE3.5 algorithm (Edson et al., 2013) was used to translate wind speeds to a standard 10 m height. This version of COARE removes the reliance on earlier ship measurements of COARE3.0 (Fairall et al., 2003) which may be subject to air-flow distortion effects, and provides better agreement with independent observations (e.g. Yang et al., 2014). COARE3.5 was also used to compute $u_*$, $U_{10n}$ ($U_{10}$ adjusted to neutral stability), bulk sensible and latent heat fluxes. The COAREG3.1 model (Fairall et al., 2011) updated earlier meteorological versions with a focus on gas transfer, and was used here to obtain predicted gas transfer velocities based on wind speed and atmospheric stability. Wave data were obtained from WaveWatch III forecast data using NOAA/NCEP winds at 0.5º (50 km) resolution, with wave parameters from these gridded data selected at the gridpoint closest to the *Tangaroa* position.





### 2.1.1 CFD airflow distortion adjustments

The airflow around RV *Tangaroa* was originally modelled by Popinet et al. (2004) using the computational fluid dynamic (CFD) model, Gerris, employing a large-eddy scale (LES) approximation. The model showed an uplift of air over the ship with an increase from ~1 m over the bow to 6 m over the beam, in addition to flow acceleration and deceleration, and the

development of turbulence and recirculation. Following Popinet et al. (2004), CFD simulations were rerun focussing on the locations of the wind instruments deployed during SOAP, to obtain airflow distortion corrections. In these large-eddy scale (LES) simulations the model did not explicitly include viscosity, but relied on numerical viscosity which in many situations produces realistic results (Popinet et al., 2004). A further limitation was that the inflow velocity was assumed uniform with height (with a slip condition at the ocean surface); thus, there is no logarithmic velocity profile with height in these results.

The numerical model of the ship's superstructure did not include the temporary foremast where instruments were mounted, and contained no detail smaller than 0.5 m (e.g. railings). In addition, no account was made for the dynamic pitch and roll of the ship.

### 2.2 Gradient flux method for DMS

The strong source of DMS in the ocean gives rise to a decreasing atmospheric concentration with height above the sea

surface in the overlying lower atmospheric marine boundary layer. Monin-Obukhov similarity (MOS) theory is used to describe the turbulent diffusion of gas away from the sea surface, with a flux, $F$, given by:

$$F \equiv -u_* C_* = -\frac{u_* \kappa}{\varphi_c(z/L)}\left(\frac{\partial c}{\partial \ln z}\right) \tag{2}$$

where $u_*$ is the friction velocity, $C_*$ is the scaling parameter for gas concentration $C$, $\kappa$ is the von Kármán constant (taken as 0.4), $\varphi_c$ is the stability function for mass, $z$ is the height above the mean water level and $L$ is the Monin-Obukhov scaling

length representing the atmospheric stability in the surface layer. Equation (2) can be integrated to give an equation expressing the concentration gradient against height:

$$C(z) = C(z_{0c}) + \frac{C_*}{\kappa}[\ln(z/z_{0c}) - \Psi_c(z/L)] \tag{3}$$

where $z_{0c}$ is the surface roughness for $C$ (McGillis et al., 2004), and $\Psi_c$ is the integrated form of $\varphi_c$ originally developed over land. The form of $\Psi_c$ has been examined over the ocean, and we follow the resulting scalar functional form used in

COARE3.5 (Edson et al., 2013). From Eq. (3) it can be seen that the measured slope of the concentration profile against ln(z) is simply $C_*/\kappa$, and this can then be used in Eq. (2), together with $u_*$, to determine the DMS flux.

In order to minimize airflow distortion effects on gas gradients, a Kiwi-Cat catamaran was instrumented, and deployed from the ship for gas sampling for periods of ~4-6 hours on 6 separate occasions (Fig. 1b) in three blooms of differing

phytoplankton species composition (see Law et al., 2017). Safety considerations limited deployments to wind speeds below ~12 m s⁻¹ (25 knots) and wave heights < 3.5 m. The ship maintained a position approximately 1 km downwind of the catamaran during sampling to avoid contamination effects, and a drogue was used to maintain the heading of the catamaran



into the wind. At the top of the catamaran mast (5.6 m above mean sea level), a small Airmar PB220 ultrasonic weatherstation measured wind speed, direction and some motion parameters. Four gas intakes, with KI oxidant scrubbers to minimize DMS loss by ozone, were fitted at logarithmic heights up the mast of the catamaran (with a mean intake height of 2.5 m). The intakes fed an autosampler that collected air simultaneously from each inlet into Tedlar or Kynaar bags for 15-

30 minutes. All heights were sampled simultaneously to avoid the possibility of gradient distortion due to non-stationarity or non-uniformity in the air-flow, which could happen if heights were sampled sequentially. Every half hour, a new set of 4 bags was sampled. At the end of each deployment the gas samples were returned to the *Tangaroa*, and analysed using sulfur chemiluminescence detection gas chromatography (SCD-GC) (Walker et al., 2016).  Calibration was carried out using an internal methylethylsulfide (MES) permeation tube for correction of detector drift, and an external DMS permeation tube

housed in a dynacalibrator.  A five-point calibration was performed twice per day, and a running standard every 12 samples (Walker et al., 2016). A subsequent international intercalibration (Swan et al., 2014) indicated that the analytical method was 93.5 ± 3.8 % accurate with 2.6 % variation. A systematic 'blank' value was removed from field samples after analysis of sample bags filled with nitrogen indicated some residual DMS in the system. The GF bag analysis assumes no loss of DMS from the bags. However, our tests revealed a 0.2 – 0.3 % decrease in DMS per hour of storage, which is somewhat higher

than that of Zemmelink et al. (2002a), who found Tedlar bags could be stored for 7 days without DMS loss. Data used here were analysed typically within 12 hours of collection so losses were minimised.

## 2.3 Eddy covariance DMS flux

The DMS eddy covariance (EC) system used during SOAP was mounted on the bowmast of the *Tangaroa* (Fig. 1a). It comprised dual Campbell CSAT3 sonic anemometers, a Systron Donner motion correction package and an air intake (Bell et

al., 2015). The EC instrumentation is located above the deck (at 12.6 m above mean sea level) and it is assumed that measurements take place in the constant flux surface layer, so that uplift should not affect the flux results. Atmospheric DMS was measured with an atmospheric pressure chemical ionisation mass spectrometer (mesoCIMS) after drying, with flux measurements made every 10 minutes, as detailed in Bell et al. (2013).  A second mass spectrometer (miniCIMS) measured flow-through sea-water DMS concentrations ($Cw$) from an intake at 6 m depth, and averaged at 5-minute intervals, with a

mean relative error of ± 5 %. A comparison of dissolved DMS was made between the miniCIMS and the SCD-GC used for the gradient measurements, which indicated a concentration offset of ~1.2 nmol L$^{-1}$ DMS (Walker et al., 2016).  For the last catamaran deployment on DOY 64, no seawater concentration or EC DMS flux data were available.

## 2.4 Spar Buoy

A spar buoy was used to provide auxiliary information on sub-surface temperature structure and turbulence during SOAP.

Three deployments were made at the initial centre of each of the three SOAP phytoplankton blooms (Law et al., 2017), lasting 1 to 4 days. The total length of the device was 6.9 m, with three vertical arms near the surface arranged around the central pole to spread buoyancy and reduce pitch/roll (Fig. 1c). The central pole extended 5.3 m below the water surface,





where a drag plate was mounted to dampen heave motion. Iridium beacons and an Airmar weatherstation were mounted on top of the spar, transmitting location, and providing wind and motion data. A vane kept the spar aligned in the direction of drift. Pitch and roll were generally < 10º (95th percentile), and the mean drift was 0.5 m s$^{-1}$. In general, the spar followed the swell motion, but rode through the shorter wind waves. Beneath the water surface, RBR TR_1060 temperature recorders,

with 0.09º C accuracy (and 0.00005º C resolution), were mounted every 0.5 m to provide information on temperature stratification.

A Vector Acoustic Doppler Velocimeter (ADV) was mounted at a mean depth of 0.7 m to provide relative current and turbulence information. The ADV had a sample rate of 16 Hz and was oriented so that the sensitive $w$ axis was perpendicular

to the drift direction and spectra would be free from vertical heave motion. In the example spectra (Fig. 2), the $w$-component in red shows a lower noise floor, and a spectral slope that is consistent with the theoretical -5/3 Kolmogorov inertial subrange, while the $v$-component has a higher noise floor and is dominated by wave motion centred around 0.2 Hz. The turbulent energy dissipation rate ($\varepsilon$) was determined using the inertial dissipation method (Drennan et al., 1996), with an extension to unsteady advection typical of a wave environment (Terray et al., 1996). This was applied to the $w$-component

across a section of the power spectral density distributions in the frequency range 1.5 to 4 Hz which was above the wave motion spectral peak. This method has provided a favourable comparison between the inertial dissipation technique using the ADV, and an Aquadopp that sampled turbulence spatially (Thomson, 2012). The ADV-derived value of $\varepsilon$ was used in the formulation of the surface renewal model (Lamont and Scott, 1970; Zappa et al., 2007) which estimates a gas transfer velocity, $K_\varepsilon$:

$$K_\varepsilon = A \, (\gamma \, \varepsilon)^{1/4} \, Sc^{-1/2} \qquad\qquad (4)$$

where $\gamma$ is the kinematic viscosity, $Sc$ is the Schmidt number, and $A$ is a constant that is likely dependent upon measurement depth.

### 3 Results

#### 3.1 Environmental conditions

During the SOAP campaign, winds varied between calm (< 0.3 m s$^{-1}$) and a maximum >20 m s$^{-1}$ during the passage of a southerly front (Fig. 3a, Law et al., 2017). For the initial part of the campaign air-sea temperature differences were small, with small heat fluxes out of the water (Fig. 3b). During DOY 51-55, warm, moist air from the north overlaid the surface ocean, resulting in a reversal of heat flux, and also a period of negative latent heat flux associated with fog. The corresponding atmospheric stability parameter (z/L) was positive (stable) during this period (Fig. 3c). Stronger winds and

colder air from the south followed, leading to greater heat fluxes out of the water and generally neutral to unstable atmospheric conditions. Significant wave heights (Fig. 3d) from WaveWatch III ranged between 2 and 5 m. However, the wave age (Cp/U$_{10}$) (Fig. 3e) indicated that swell dominated the wave field for the majority of the time, which is typical of



this region (Smith et al., 2011). This indicated that the most energetic waves were travelling significantly faster than the wind speed, and were undergoing active development only during occasions of rapidly increasing winds. An example of the latter was on DOY 60.7 when a storm front crossed the region, although the actively wave-generating systems generally passed relatively rapidly. During the catamaran deployments examined here, wind speeds ranged between 4 and 10 m s$^{-1}$,

with mean values shown in Table 1. The average wave height varied between 1.3 and 2.2 m, with a wave age well above 1.2, indicating swell-dominated wave conditions.

**3.2 Gradient Fluxes**

Examples of the atmospheric DMS profiles obtained on the catamaran are shown in the upper panels of Fig. 4. Where possible, replicate gas chromatograph analyses were carried out on each sample bag. When multiple analyses are available

the standard deviation of these are plotted as error bars in the upper panels of Fig. 4, and are typically on the order of ±5 ppt. The lower panels of Fig. 4 show DMS versus the log of the height coordinate corrected for atmospheric stability, together with a least square fit of a logarithmic profile (shown in red). From Eq. (2) and (3), the DMS flux is related to this slope. The quality of the profiles varied, as shown in the two examples in Fig. 4, due to both instrumental and environmental causes. Data were only used where the gradient fit $r^2$ value exceeded 0.5. The residual error of the fit was used to calculate the error

in parameters, which in turn was used to determine the error bars for each sampling point. During deployment 4 on DOY 53, the upper level intake was contaminated and not used in the analysis. As the campaign progressed, improvements were made to the technical method, primarily autosampler control for bag collection, which resulted in improvements in the quality of the concentration profiles, and reduced error bars.

Since $u_*$ was not directly measured on the catamaran, there were two alternatives for obtaining it for calculation of DMS fluxes from Eq. (2): the bulk flux (corrected for airflow distortion) from the nearby RV *Tangaroa*, or the directly measured, motion-corrected $u_*$ from EC measurements onboard *Tangaroa*. The gradient fluxes estimated using the EC $u_*$ (Landwehr et al., 2017) are shown in Fig. 5a in red, overlain on the EC continuous 10-minute data, for 20 days of the SOAP voyage. Figure 5a highlights the wide range in fluxes during SOAP, reflecting the variability in wind speed, and the spatial and

temporal variability of aqueous DMS concentration. The EC data encompasses the full range of data, including when the ship was on station and underway, and traversing diverse DMS source regions, whereas the comparisons of EC and GF data in Fig. 5b-f are when the ship was on-station and so variability was lower. Nevertheless, the DMS flux can vary by a factor of 8 at a fixed location during the day, as on DOY 49 (Fig. 5e).

Despite the overall variability of the flux, the eddy covariance and gradient flux measurements agree very well overall (Fig. 5b-f), particularly when considering that their spatial separation varied between 300 m and 13 km (Table 1). An exception to this was DOY 48, when there was believed to be a leak in the autosampler, and so data from DOY 48 are not used in subsequent analysis. In panels 5b-f the error bars are calculated from the least squares fit of the gradient, as in Fig. 4,





combined with the relative standard deviation of $u_*$ during the sampling period, thus indicating both the quality of the gradients, and the variability in winds speed over the sample. The best agreement was obtained on DOY 53 and 58, under conditions of light and moderate wind respectively. On DOY 60 the *Tangaroa* moved up to 13 km away from the catamaran in order to retrieve other instruments which may account for some of the difference. At times (e.g. DOY 48) flux variability

on a timescale of hours can be O(100 %), while on other occasions (e.g. DOY 49) consistent trends are evident on this timescale.

A direct comparison between the GF and EC fluxes is shown in Fig. 6, with EC and GF fluxes averaged over the same 15 – 30 minute intervals. The GF values using $u_*$ derived from the motion-corrected EC system provide better agreement than u*

derived from bulk fluxes incorporating airflow corrected wind speeds. The least squares linear fit has a slope of 0.96 ($r^2$ = 0.89), while the bulk formula gives lower flux values by an average of 10 %, with a fitted slope of 0.78 ($r^2$=0.86). This suggests that the CFD modelling has over-compensated for the airflow acceleration, resulting in a $u_*$ that is too low, as examined in detail by Landwehr et al. (2017). It is likely that small-scale features of the *Tangaroa*, such as the foremast and railings, which were not included in the model, may have a significant effect on the airflow corrected wind speeds. In this

case, both the airflow distortion and the bulk formulation have increased the uncertainty in gradient fluxes.

### 3.3 Comparison of GF and EC gas transfer velocities

The commonly used equation for bulk gas transfer, Eq. (1), can be used to derive the gas transfer velocity $K$ from the measured flux, and water and air DMS concentrations:

$K = F/(Ca/H - Cw)$                          (5)

$K$ was obtained from the measured DMS fluxes from both EC and GF methods, and the DMS concentration in air ($Ca$) and water ($Cw$). The 10-minute EC values of $K$ (Bell et al., 2015) were again averaged over the same time intervals as the GF values. $K$ values are often presented as an average over several days or within wind speed bins, due to the inherent or instrumental variability; however we consider much higher temporal scales, with estimates of $K$ over 30-minute intervals. Figure 7a shows a plot of $K$ derived by both methods where coincident data were available, against wind speed adjusted to

neutral stability, together with the COAREG3.1 parameterization. The GF $K$ values are generally lower than the EC $K$ values, consistent with a similar bias in the flux comparison (Fig. 6), but show better agreement with the COAREG curve than EC. However, there is a clear anomaly with the COAREG curve for the period of low winds on DOY53, despite good agreement between GF and EC methods at this time.

In Fig. 7b the GF $K$ data are colour-coded with respect to atmospheric stability and U$_{10}$ is *not* adjusted to neutral stability. COAREG $K$ is shown for the mean conditions during SOAP, and also for extremes of air-sea temperature difference during SOAP (thin lines). The relative humidity extremes during SOAP were also used in this calculation. The range in the COAREG curves due to atmospheric stability effects is relatively small in comparison to the experimental errors in the GF $K$



data, which mostly fall within error bounds of the curve.    However, it is clear that the anomalous values on DOY53 occurred under conditions of strong atmospheric stability that are outside the uncertainty bounds from COAREG.

**3.4 Ocean Stratification**

The spar buoy monitored temperature stratification in the upper 5 m of the water column during the 3 deployments. Nocturnal temperature profiles were well mixed in this depth range by convection, whereas minor daytime stratification developed in the early afternoon. However, on two days, strong stratification up to $1^o$C m$^{-1}$ developed.  The first period on DOY 47 developed under conditions of extremely light winds (<3 m s$^{-1}$) and high insolation (Fig. 8a). The solar irradiance peaked at approximately DOY 47.05 at this longitude, with the build-up in stratification peaking later at DOY 47.12, and maximum warming occurring in the upper 2 m. Under these calm conditions, the air temperature also increased by $3^o$C during the afternoon.  Anomalously high DMS fluxes followed this, and Walker et al. (2016) suggest that the stratification led to an accumulation of DMS in near-surface waters, which ventilated to the atmosphere when winds subsequently increased.

A second period of strong ocean stratification was measured on DOY 53 under somewhat stronger wind speeds ~5 m s$^{-1}$. The stratification on DOY 53 was associated with an extended period of atmospheric stability with reversed atmospheric heat fluxes when warmer moist air overlaid the ocean (Fig. 3b). It was during this period that the strong discrepancy between measured $K$ values and COAREG was observed (Fig. 7b), with significantly lower derived $K$ values from EC and GF than for COAREG.  The temperature structure of the upper 4.3 m of the water column (Fig. 8b) shows strong stratification, but structured in quite a different manner to DOY 47. During DOY 53, stratification extended well below the depth of the spar buoy. At times temperature oscillations also occurred throughout the depth of the spar buoy, with an amplitude that decreased towards the water surface, consistent with internal wave activity with a period of ~12 min. These increasing temperature excursions with depth are in contrast to DOY 47, where maximum temperatures were confined to the upper 2 m where solar insolation was strongest. While the amplitude of the internal wave was damped near the surface, it potentially contributed to surface convergence/divergence zones and surface patchiness. This process may have contributed to the heterogeneity of surface DMS concentrations reported by Bell et al. (2015).  The stratification driven by the reversed heat flux persisted until after midnight local time.

**3.5 Turbulent dissipation rate and $K$**

Turbulence derived from the ADV was measured during the 3 deployments of the spar buoy during the SOAP campaign. The spar buoy drifted with the ocean currents, with a track marked by characteristic inertial oscillations. The position of the ship was dictated by multidisciplinary measurement requirements, so that the distance between the ship and spar buoy was ~15 km during catamaran deployments on DOY 47 and 53; at other times ship-based activities took place 25 – 45 km away.





Wind records from the spar buoy suggest that this was close enough to be within the same mesoscale meteorological regime, but far enough that some differences in wind speed were apparent (e.g. Fig. 8a).

$K_\varepsilon$ was derived from water-side turbulence using Eq. 4 and $A$=0.2, and is compared with the COAREG $K$ and $K$ calculated
from EC (Fig. 9). Whereas $K_\varepsilon$ is derived directly from turbulence measurements, the main input to the COAREG parameterization is wind. The lowest values of $\varepsilon$ were observed during the calm, stratified period on DOY 47 when turbulence was expected to be weakest, and DOY 53 when strong waterside stratification was observed. However, $K_\varepsilon$ did not fall off to the same extent as $K$ from COAREG or EC under light winds on DOY 47 and 55. Within the limited data coverage, $K_\varepsilon$ follows the trends of the COAREG $K$ but there is noticeable deviation from EC data on DOY 48 which started
late on DOY 47, when EC fluxes were significantly higher. This was also the period when EC exceeded COAREG, and significant DMS enhancement in the surface microlayer was reported by Walker et al. (2016). These observations support the suggestion that the enhanced DMS fluxes on DOY 48 were associated with biogeochemical effects rather than physical processes driven by the wind.

### 4 Discussion

To our knowledge this is the first published study to directly compare EC and gradient flux methods for DMS from a seagoing platform. The results show good agreement between these two independent methods (Fig. 6), giving weight to the validity of both sets of measurements. The slope of the regression between the two methods (0.96) and $r^2$ value (0.89) shows considerably better agreement than a previous comparison of GF with REA (Hintsa et al., 2004), which found a difference of a factor of 2. A limitation of REA over the ocean is that it requires real-time motion correction, which is extremely
challenging as post-processing corrections like those used in EC (e.g. incorporation of time symmetric filtering of ship motion) cannot be made. In contrast to the Hintsa et al. study, the EC and GF techniques also showed good agreement under conditions of strong atmospheric stability (Fig. 7). Despite this agreement, there were significant differences from the $K$ predicted by the COAREG model during the stratification event on DOY 53. The following discussion considers the relative merits of the GF and EC techniques, and assesses the factors that may give rise to variability between these techniques, and
also differences with COAREG model outputs. The application of turbulent dissipation rate is also considered.

### 4.1 Logistical considerations

The GF method has some benefits for flux determination. The accumulation of sample gas over 15-30 minutes followed by later off-line analysis allows good analytical precision by standard gas chromatographs and so instrument costs are modest, compared to EC which requires fast response, high-precision gas sensors. The use of a catamaran for sampling also has
advantages in that it is not influenced by the major air flow distortion of a large ship, and air sampling is close to the water surface where gas vertical gradients are largest, so improving sensitivity. On the other hand, deployment of the GF technique





on a remote platform such as a catamaran, is logistically more difficult, and limited to a smaller range of open ocean conditions, wind speeds less than 10 m s$^{-1}$ and moderate wave conditions. EC is not limited in this way, and is able to provide a more detailed dataset across a full range of conditions, including high wind speeds where data is sparse.

**4.2 Assumption of Monin-Obukhov similarity (MOS)**

One of the fundamental issues with the GF method is a dependence on assumptions of Monin-Obukhov similarity (MOS). MOS is well established over land, but there has been some caution in translating MOS theory concepts to the open ocean which is complicated by an actively deforming surface that may generate a wave boundary layer (WBL) just above the ocean surface in which a wave influence may exist. However, there have now been sufficient studies (e.g. Edson et al., 2004; 2013)
that provide compelling evidence of the successful application of stability functions developed over land to neutral and unstable conditions at heights above the wave boundary layer. The height of the WBL has been subject to many definitions, from z ~ Hs, the significant wave height (Edson et al., 2004), to 3.7 Hs (Chalikov, 1995). It seems that most interaction between waves and fluxes occurs within ~1 Hs. This would severely limit the ability of the EC technique to determine u* when used within the WBL, due to wave-generated pressure and vertical motion. With the GF technique, we have taken the
$u_*$ component from the ship-based EC measurements, which are assumed to be above the WBL. In contrast, gas concentrations are not subject to wave pressure effects, and consequently it has been suggested that wave modulation of gas fluxes would be less than for momentum fluxes (Edson et al., 2004). It is worth noting that Prytherch et al. (2015) found a residual motion signal in momentum flux spectra, which was caused by motion-induced flow distortion rather than wave-induced momentum flux. Their results suggest that WBL effects may not be as severe as has been suggested. Indeed, the
consistency between gradient fluxes measured close to the water surface and EC fluxes measured at 12 m height, and averaged across many wave cycles (see Fig. 5) suggests that any WBL effect was not large in the SOAP study region, which was dominated by ~2 m swell (Fig. 3c, Table 1). We recommend that the $u_*$ used in the GF calculation is obtained from ship-based measurements above the influence of the WBL, but within the constant flux surface layer, as long as the ship is close by during the 15-30 min measurement interval. In the absence of EC measurements on the ship the alternative is to use a
bulk parameterization of $u_*$ based on measured wind speed; however, this may be subject to significant airflow distortion which numerical CFD airflow correction cannot always fully resolve (Fig. 6; Landwehr et al., 2017) resulting in GF estimates that do not compare as favourably with the EC fluxes (Fig. 6, this paper).

A further consideration is the extent that the GF concentration profiles were affected by the heave of the catamaran following
the wave motion. The catamaran will ride through short period waves, but follow the longer waves. In the hypothetical case where the concentration profiles remain unchanged, the catamaran autosampler collects air while the intake traverses air over a range of heights as the catamaran follows the wave motion. We have evaluated the impact under the assumption of a logarithmic concentration profile with height, and an equal time spent over the spread of heights. The log profile was



integrated over the height of a typical wave to find the average concentration and its difference from an undisturbed measurement. The integration has a simple analytic solution under the assumption of a sinusoidal vertical displacement due to a monochromatic swell. For typical measured log profiles, the effect of a peak-to-trough wave height up to 2 m was less than 1 % at the upper three measurement heights. For the lowest height (0.5 m), a 1 m wave height will result in a measured

increase on the order of 2 % over the undisturbed value. This is likely to represent a worst case, since the concentration profile close to the water surface will be uplifted to some degree by the wave progression, along with the catamaran (Mahrt et al., 2005; Prytherch et al., 2015). Consequently, this effect is regarded as relatively minor in comparison with other sources of uncertainty. This provides a further contrast to EC, in which the significant correlation of gas concentration with vertical velocity induced by wave motion has to be carefully compensated for.

**4.3 Footprint difference**

There is a significant difference in the sampling footprint between the gradient measurements on the catamaran close to the water surface centred at 2.5 m, and the EC measurements based at 12 m on the ship. The footprint model of Kljun et al. (2015) predicts that the distance to peak footprint will be over twice as large on the *Tangaroa* as the catamaran. The footprint model assumes a spatially homogenous source region. However, Bell et al. (2015) showed that strong point sources typical

of a bloom can influence the footprint from much larger distances than a uniform source. In particular, the EC footprint has the potential to be influenced by seawater DMS concentrations up to ~5 km upwind (depending on wind speed). In contrast, the GF footprint at similar wind speeds only extends ~ 2 km upwind. Hence, some difference in fluxes between EC and GF can be expected when the seawater DMS distribution is non-uniform. There were factors that mitigate against the footprint difference during SOAP. First, the catamaran was always positioned upwind of the ship by ~2 km to avoid exhaust

contamination (Table 1), improving the footprint match. Second, both catamaran and ship were subject to a slow (0.5 m s$^{-1}$) wind-driven drift through water, which over the course of 30 min would provide close to a kilometer further spatial averaging of any non-uniformity in seawater DMS.

**4.4 Boundary layer stability and near surface water stratification**

Our understanding of fluxes under stable conditions is much more limited than neutral or unstable conditions (Edson et al.,

2007), and data from stable conditions are often simply discarded (e.g. Yang et al., 2011). It is perhaps surprising that the transfer velocity results during the stable conditions on DOY 53 were so tightly clustered (Fig. 7), and showed such close agreement between GF and EC, since Blomquist et al. (2010) report that the theoretical flux uncertainty (due to atmospheric turbulence) increases markedly under stable conditions. Both GF and EC values of $K$ were consistently lower than the COAREG parameterization. The downward heat flux in these stable situations supresses the vertical motion of shear-

generated atmospheric turbulence, which is reflected in the stability functions ($\Psi_c$). The agreement between the GF method (dependent upon $\Psi_c$) and EC (independent of $\Psi_c$), suggests that the discrepancy with COAREG does not arise from the stability functions themselves. DMS exchange is predominantly water-side controlled, so it is also important to determine





the characteristics of the water column, where conceptual models assume that the surface interface is freely accessible to renewing/replenishing eddy activity from below. The downward heat flux during SOAP caused stratification of near-surface waters (Fig. 8b), limiting mixing and near-surface turbulence, and the surface mixed layer depth shoaled from 21 m to 14 m (Law et al., 2017). Sims et al. (2017) have also shown that near surface stratification can lead to gradients in gas

concentrations. The stable ocean-side also provided conditions to support internal waves which are likely to have contributed significant patchiness in the ocean distribution of seawater DMS through the convergence/divergence of the wave motion.

On the other occasion when strong ocean stratification was observed (DOY 47, Fig. 8a), observed values of $K$ were also lower than expected from COAREG (Fig. 9). Indications of microlayer enrichment followed these calm conditions when

strong near-surface stratification had built up (Walker et al., 2016). It is often assumed that at low wind speeds, transfer will be underpinned by convectively generated turbulence. In contrast, the conditions here produced stable ocean stratification during the day near the surface, which suppressed gas transfer. Under the low wind speed on DOY 47, it may be more accurate to model the sea-surface as a rigid surface which would have a Schmidt number exponent of 2/3, rather than the usual wave roughened surface (Schmidt number exponent = 0.5). Accounting for this would raise the $K_{660}$ values by 10 %,

but this is still much lower than is required to match the COAREG model.

The stratification driven by downward heat flux related to mesoscale conditions is likely to be more persistent than daytime solar heating under calm conditions. During SOAP these downward heat flux conditions lasted 3 days, so could have a significant effect on net DMS fluxes. The concentration of material near the sea surface may have led to enhanced near

surface production of DMS and elevated concentrations. Underway measurements, which are made using a typical intake 5-6 m beneath the surface, will undersample such a surface concentration, causing a bias in the estimation of $K$.

### 4.5 Turbulent Energy Dissipation Rate

Models of gas exchange suggest that the turbulent energy dissipation rate ($\varepsilon$) is a parameter more directly related to $K$ than wind speed, since it is generated by both wind stress and breaking waves. While $\varepsilon$ has been used to estimate gas transfer

velocities in several estuarine and coastal measurements (e.g. Zappa et al., 2007; Tokoro et al., 2008), observations of the open ocean relationship with $K$ are limited to just one study (Esters et al., 2017). The results here show that $K_\varepsilon$ derived from $\varepsilon$ followed the COARE parameterization more closely than the EC data did, but it did not exhibit the same range of variation, particularly during lulls in wind speed. In particular, while minimum values of $\varepsilon$ were observed during stratification events, $K_\varepsilon$ did not decrease to such low levels as the observed EC or COAREG $K$. This may be due in part to

the spatial separation of spar buoy and ship, since wind speed at the spar buoy did not drop to the same extent as the ship. The ADV was orientated on the spar to minimize interference by turbulence from waves impacting the spar structure, but this may not have removed all of the influence. Therefore, the spar buoy estimates of $\varepsilon$ are likely to be an upper limit for the true value. Measurement of $\varepsilon$ from a ship would clearly not be possible due to much larger wave and wake effects.





The SOAP spar buoy turbulence measurements were taken at a mean depth of 0.7 m. This is very close to the surface, but the actual exchange happens at the air/sea interface where measurements are extremely difficult to obtain. Some studies have attempted to extrapolate to the surface (Esters et al., 2017) while others (e.g. Zappa et al., 2007) have not. This will affect the

constant $A$ in Eq. 4. Breaking waves generate subsurface turbulence which may exceed the wind stress generated turbulence (following a 'law of wall' variation with depth) by an order of magnitude (Terray et al., 1996). Extrapolation to the surface is thus dictated by the parameters of the model deployed to parameterize this effect. Terray et al. (1996) found enhancement of turbulence within a layer ~1 $Hs$ deep, but within this layer it was roughly constant. By this measure, and with wave height >1 m at all times (Fig. 3), the ADV at 0.7 m depth was within the enhanced turbulence zone. Thus, keeping $A$ fixed is

appropriate. The caveat to this is that during SOAP the $Hs$ was dominated by swell rather than an actively breaking young sea, which would have elevated turbulence levels.

A further factor that may affect the parameterization of $K$ using $\varepsilon$ is that recent laboratory measurements by Deane et al. (2016) find that turbulence dissipation near the breaking wave crest is saturated, and does not vary much with wavelength and slope. They suggest that either bubbles limit the degree of turbulence intensity, or that the turbulence is spread across

varying depths. In the latter case, $\varepsilon$ would need to be measured at multiple depths and integrated through the water column. While appealing from a physical point of view, the utility of using $\varepsilon$ in a predictive sense for $K$ requires further testing and confirmation, particularly in view of the weak dependence of $K_\varepsilon$ on $\varepsilon$  (i.e. $K_\varepsilon \sim \varepsilon^{1/4}$).

**5 Conclusions**

Sea-to-air DMS fluxes measured above the ocean exhibit significant scatter due to the heterogeneity of seawater DMS in the horizontal (Bell et al., 2015) and vertical (Sims et al., 2017), potential microlayer influences (Walker et al., 2016), and also the inherent stochasticity of the turbulent atmospheric boundary layer. Fluxes have been measured successfully in the terrestrial environment by both eddy covariance (EC) and gradient flux (GF) techniques, but the marine environment poses

much greater challenges with motion corrections, air-flow distortion, and aerodynamic complications caused by a dynamically disturbed water surface. The SOAP campaign provided a unique opportunity to directly compare the EC and GF methods. The GF was deployed on an independent platform from the ship with minimal air flow distortion. Despite the differences in platforms, the two techniques showed good agreement (regression slope = 0.96, $r^2$ = 0.89), providing support for the validity of both techniques in this environment. The range of variability of fluxes over periods of 4 – 5 hours on

occasions greatly exceeded the experimental uncertainties. The use of turbulent eddy dissipation rates near the surface to calculate gas transfer velocity ($K_\varepsilon$) was also trialled, using measurements from a drifting spar buoy.  This method showed closer agreement with COAREG than EC in general, although it did not reflect the range of variation observed with the other



techniques. During a period of atmospheric and near surface ocean stratification, EC and GF agreed well, but were significantly lower than predicted by the COAREG parameterization. These data suggest a systematic discrepancy under stable conditions and is a further illustration that factors other than wind speed are important for air/sea gas fluxes. The SOAP observations have provided valuable insight into the factors modulating gas transfer under stable conditions, for which there is less understanding than for neutral or unstable conditions.

*Data availability.* The underway DMSsw can be downloaded at http://saga.pmel.noaa.gov/dms/select.php. The remaining data are available by request email to m.smith@niwa.co.nz.

*Competing interests.* The authors declare that they have no conflict of interest.

*Special issue statement.* This article is part of the special issue "Surface Ocean Aerosol Production (SOAP) (ACP/OS interjournal SI)". It is not associated with a conference.

### 6 Acknowledgements

The authors thank the Master, officers and crew of the RV *Tangaroa* for making this field work possible, Fiona Elliott, Steve George, Andrew Marriner and John McGregor for invaluable assistance with the catamaran and spar buoy work, and Craig Stewart and Craig Stevens for the design of the spar buoy. We also thank Sebastian Landwehr for the eddy covariance $u_*$ data, and Warren de Bruyn, Christa Marandino and Cyril McCormick for the UCI CIMS measurements. This research was supported by funding from NIWA's Climate and Atmosphere Programme 3. CIMS flux measurements were supported by the NSF Atmospheric Chemistry Program (grant nos08568, 0851472, 0851407 and 1143709) as a contribution to US SOLAS.



**Tables**

| Deployment # | DOY (UTC) | Time (NZST) | Mean wind speed (m s⁻¹) | Ca (ppt) | Mean Cw (nM) | Mean flux (umol m⁻² d⁻¹) | Wave Hgt (m) | Wave Age | Distance ship-cat (km) |
|---|---|---|---|---|---|---|---|---|---|
| 1 | 48.08-48.23 | 17-Feb 14:00-18:30 | 5.9 | 300-500 | 16.5 | 10 | 1.7 | 3.2 | 0.3-3 |
| 2 | 49.08-49.3 | 18-Feb 14:30-18:00 | 6.0 | 350-750 | 17.0 | 30 / 15-45 | 1.6 | 3.4 | 1-3 |
| 3 | 53.10-53.23 | 22-Feb 14:30-18:00 | 4.3 | 285-400 | 13.9 | 5 | 1.9 | 4.0 | 1-3 |
| 4 | 58.06-58.25 | 27-Feb 13:30-17:00 | 7.7 | | 6.4 | 11 | 2.0 | 2.6 | 0.7-2 |
| 5 | 60.06-60.21 | 29-Feb 13:30-17:00 | 7.2 | 175-340 | 5.0 | 10 | 2.1 | 2.3 | 0.6-13 |
| 6 | 64.85-65.0 | 5-Mar 8:30-12:00 | 3.6 | 40-210 | NA | 7.5 | 3.3 | 5.6 | 1-2.6 |

Table 1. Summary of environmental conditions and DMS concentrations in air (Ca) and water (Cw) during Gradient Flux measurements.



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





**Figures**

Figure 1.
a) Positions of EC sampling mast and AWS meteorological anemometer on RV *Tangaroa*    b) Catamaran gradient flux sampling platform with intakes up the mast   c) Spar buoy with temperature loggers at 0.5 m
intervals and ADV close to sea surface

Figure 2
Example of Vector velocity spectra, at a mean depth of 0.7 m, for the 3 components. The inertial dissipation method was applied to the w component across the frequency range 1.5 to 4 Hz, shown by vertical dashed
lines. The black dashed line indicates the slope of the theoretical Kolmogorov inertial subrange.

Figure 3.
Background environmental parameters. a) Wind speed adjusted to 10 m height (U10) b) Sensible and latent heat fluxes derived from COARE3.5 c) Stability parameter z/L, with negative denoting unstable conditions; zero,
neutral stability; and positive, stable conditions d) Significant waveheight from WaveWatch III e) Wave age ($C_p/U_{10}$) with 1.2 denoting full development. Spar buoy deployments are denoted by the red horizonal bars, and catamaran GF deployments by the * symbol in the top panel.

Figure 4.
Examples of DMS gradients a) good quality data b) poorer quality.  The upper plots show the decrease in DMS concentration away from the sea surface. The lower plots show the concentration decrease against the log of stability adjusted height, as described in Equation 3, with the least squares fit of the log profile shown in red.

Figure 5.
DMS fluxes during SOAP, estimated by eddy covariance (EC, blue dots), and gradient flux (GF, red dots) during catamaran deployments for a) all voyage data and b)-f) individual periods of catamaran deployment, using EC u*. Note the change in scale for d) – f).

Figure 6.
Direct comparison of DMS flux measured by GF and EC techniques, using different evaluations of u* for GF, from  direct EC measurements (black filled circles), and from bulk formulae (open circles).

Figure 7.
a) DMS gas transfer velocities from GF and coincident EC as a function of wind speed adjusted to neutral
stability ($U_{10n}$). The transfer velocity from the COAREG3.1 algorithm calculated for neutral stability with parameters A=1.6; B=1.8, is shown by the green line. b) GF transfer velocities as a function of $U_{10}$, colour coded by atmospheric stability (z/L). The transfer velocity from COAREG for mean conditions, together with the extremes of stability are shown by the green lines, unstable (upper) and stable (lower).  The cluster of points with positive stability with low k660 are from DOY53.



Figure 8
Wind speed (upper) and subsurface temperature structure measured by the spar buoy (lower)  a) under calm atmospheric conditions on DOY 47 (following Walker et al., 2016)  b) during stable atmospheric conditions with downward heat flux on DOY 53. Indications of internal wave activity (period 12 min) are centred at 53.15.

5  Depths of temperature measurements (m) are shown on the legends.

Figure 9.
Gas transfer velocity, K, calculated from turbulent eddy dissipation rate  ($\varepsilon$), COAREG, and EC 10-min data.





**Figure 1.**
a) Positions of EC sampling mast and AWS meteorological anemometer on RV *Tangaroa*
b) Catamaran gradient flux sampling platform with intakes up the mast
c) Spar buoy with temperature loggers at 0.5 m intervals and ADV close to sea surface

(a)

AWS

DMS Eddy covariance mast (UCI)

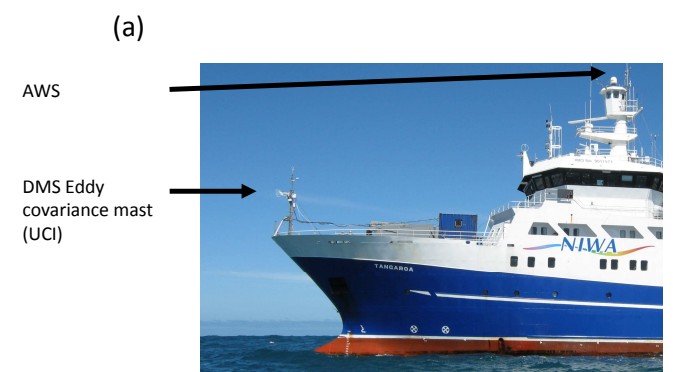

(b)

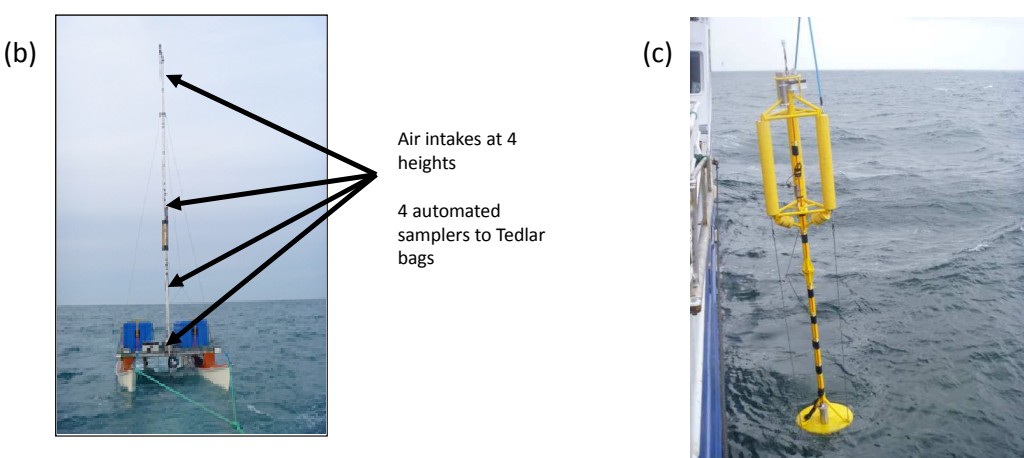

Air intakes at 4 heights

4 automated samplers to Tedlar bags

(c)





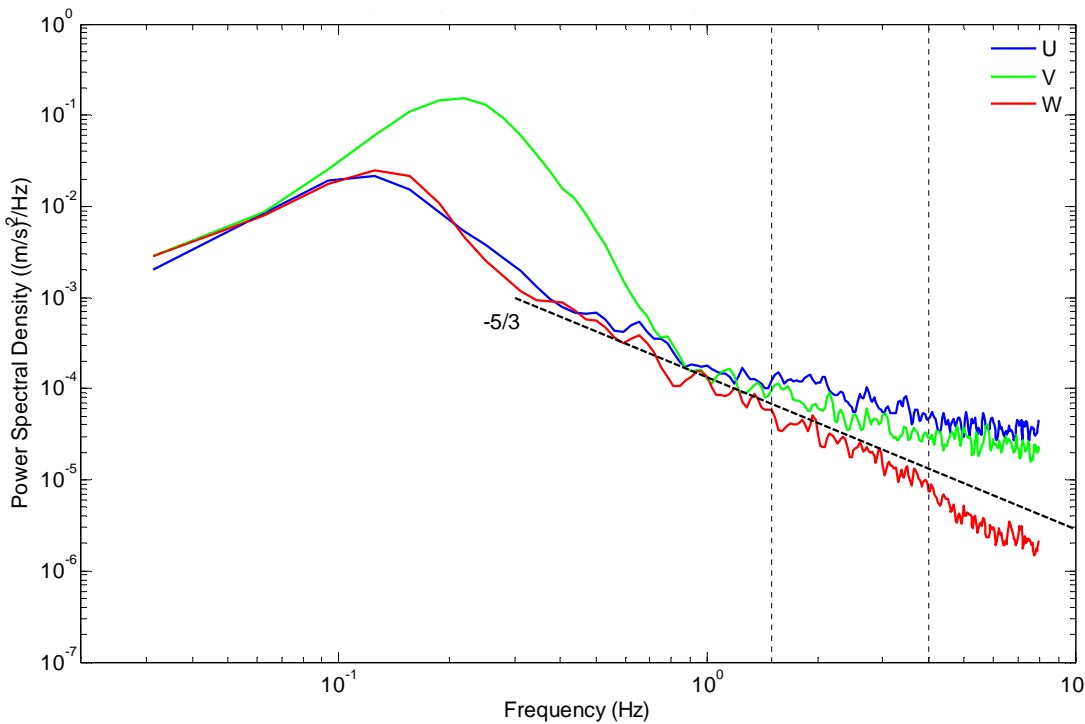

**Figure 2.** Example of Vector velocity spectra, at a mean depth of 0.7 m, for the 3 components. The inertial dissipation method was applied to the w component across the frequency range 1.5 to 4 Hz, shown by vertical dashed lines. The black dashed line indicates the slope of the theoretical Kolmogorov inertial subrange.




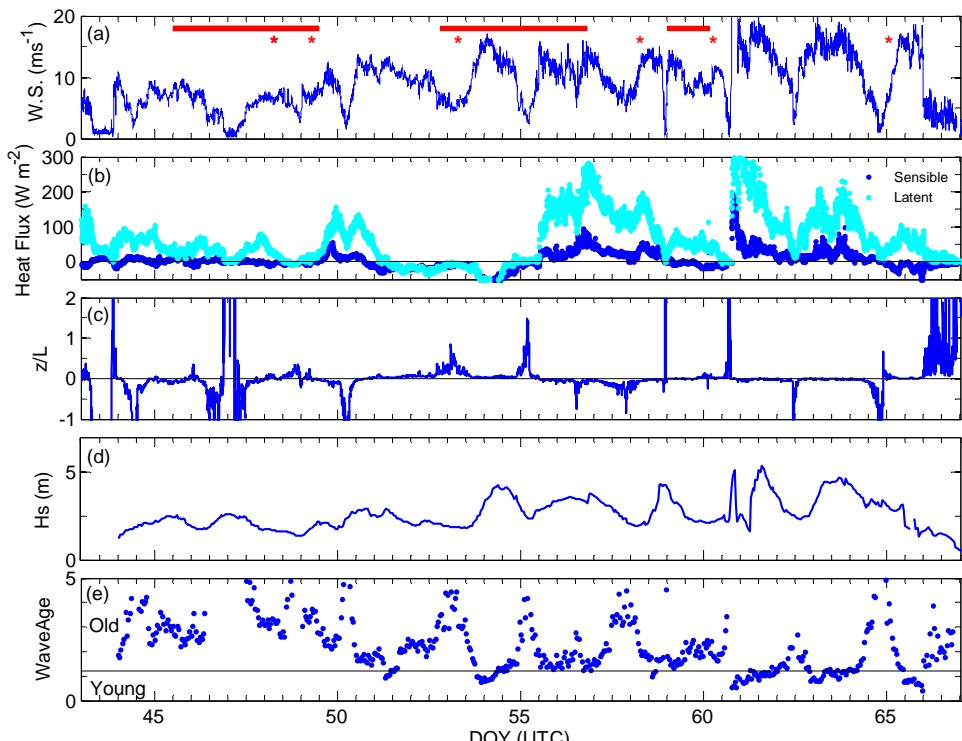

**Figure 3.** Background environmental parameters. a) Wind speed adjusted to 10 m height (U10) b) Sensible and latent heat fluxes derived from COARE3.5 c) stability parameter z/L, with negative denoting unstable conditions; zero, neutral stability; and positive, stable conditions c) significant waveheight from WaveWatch III d) Wave age $(Cp/U_{10})$ with 1.2 denoting full development. Spar buoy deployments are denoted by the red horizontal bars, and catamaran GF deployments by the * symbol in the top panel.




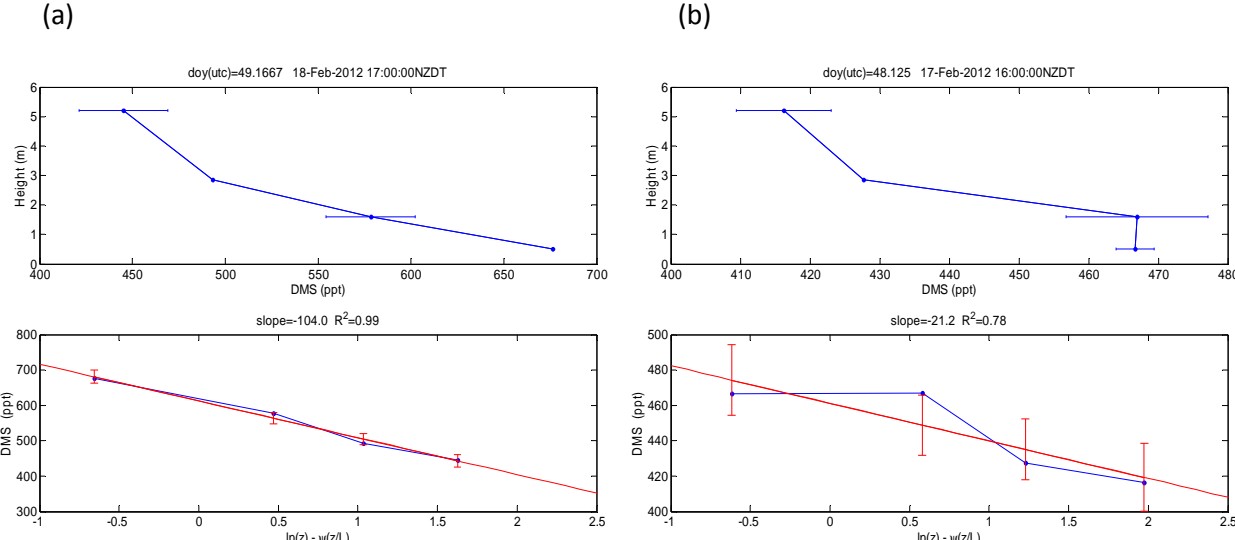

**Figure 4.** Examples of DMS gradients a) good quality data b) poorer quality. The upper plots show the decrease in DMS concentration away from the sea surface. The lower plots show the concentration decrease against the log of stability-adjusted height, as described in Equation 3, with the least squares fit of the log profile shown in red.





**Figure 5:** DMS fluxes during SOAP, estimated by eddy covariance (EC, blue dots), and gradient flux (GF, red dots) during catamaran deployments for a) all voyage data and b)-f) individual periods of catamaran deployment, using EC u*. Note the change in scale for d) – f).

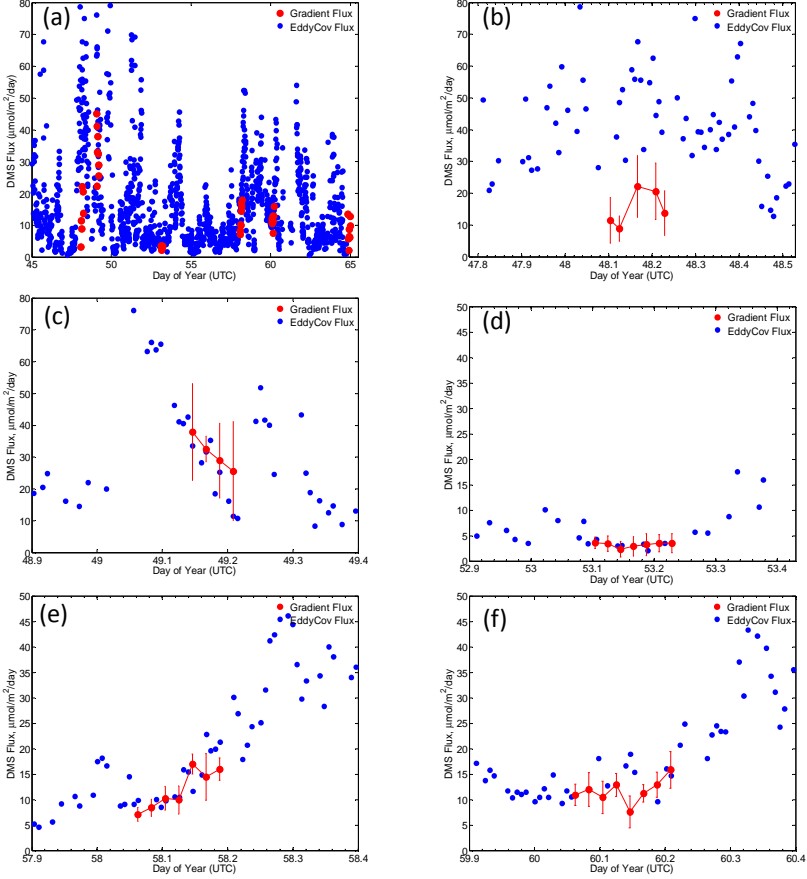



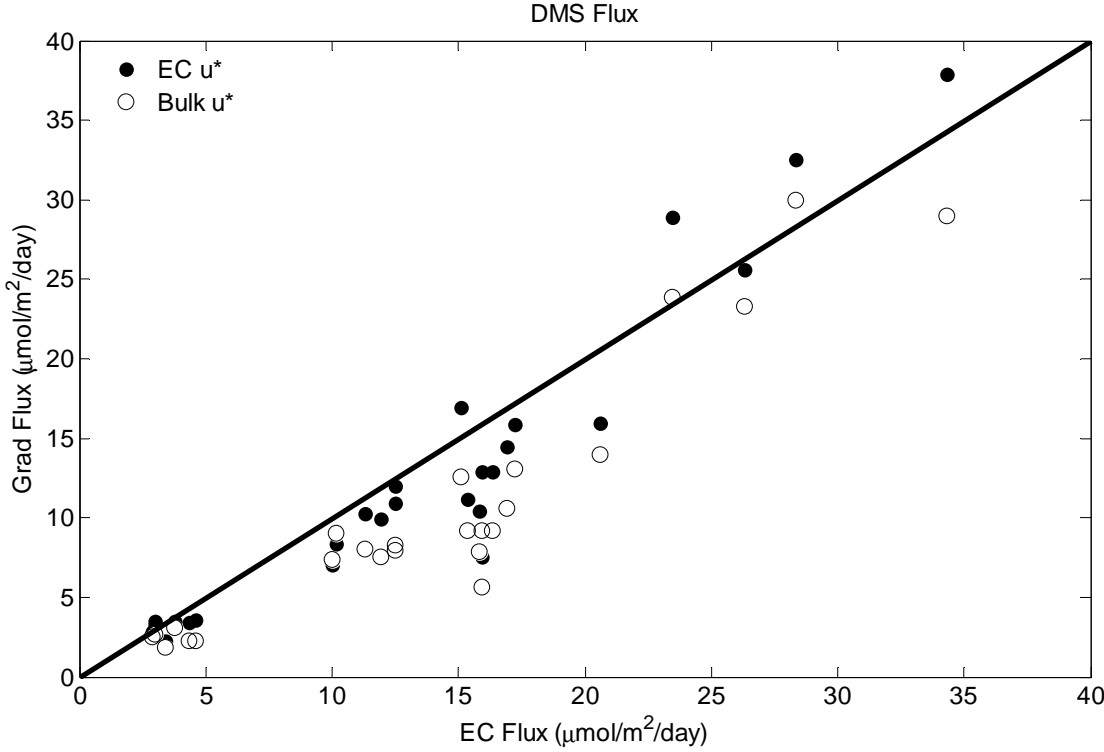

**Figure 6.** Direct comparison of DMS flux measured by GF and EC techniques, using different evaluations of u* for GF, from direct EC measurements (black filled circles), and from bulk formulae (open circles).




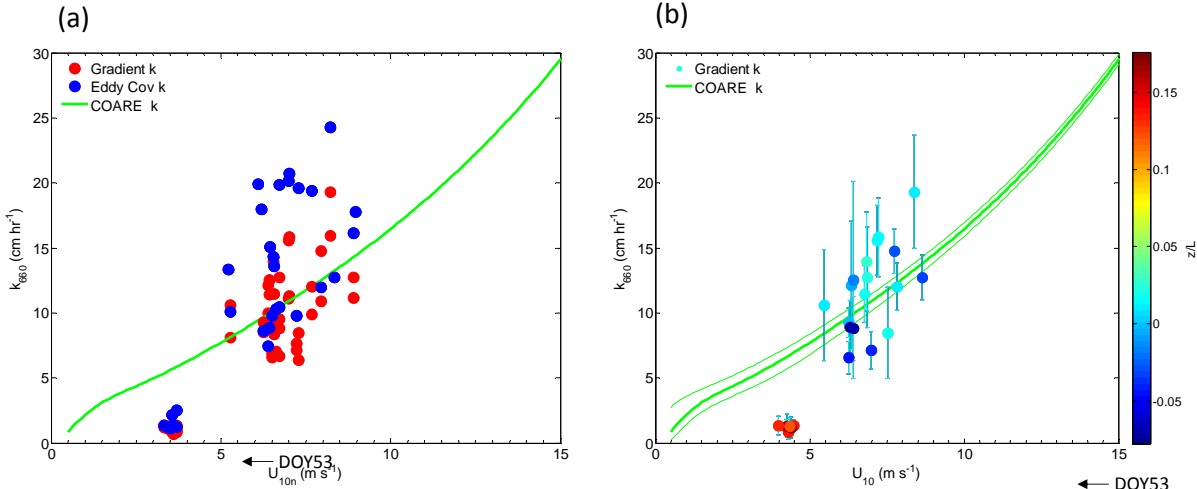

**Figure 7.** a) DMS gas transfer velocities from GF and coincident EC as a function of wind speed adjusted to neutral stability ($U_{10n}$). The transfer velocity from the COAREG3.1 algorithm calculated for neutral stability with parameters A=1.6; B=1.8, is shown by the green line. b) GF transfer velocities as a function of $U_{10}$, colour coded by atmospheric stability (z/L). The transfer velocity from COAREG, together with the extremes of stability are shown by the green lines, unstable (upper) and stable (lower). The cluster of points with positive stability with low k660 are from DOY53.



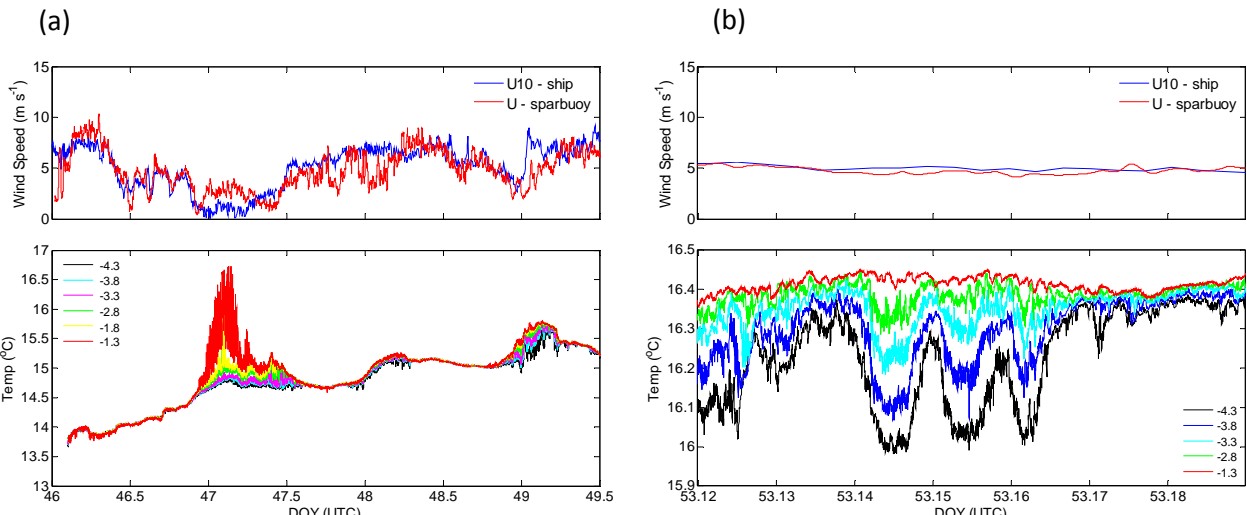

**Figure 8.** Wind speed (upper) and subsurface temperature structure measured by the spar buoy (lower) a) under calm atmospheric conditions on DOY 47 (following Walker et al., 2016) b) during stable atmospheric conditions with downward heat flux on DOY 53. Indications of internal wave activity (period 12 min) are centred at 53.15. Depths of temperature measurements (m) are shown on the legends.

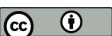


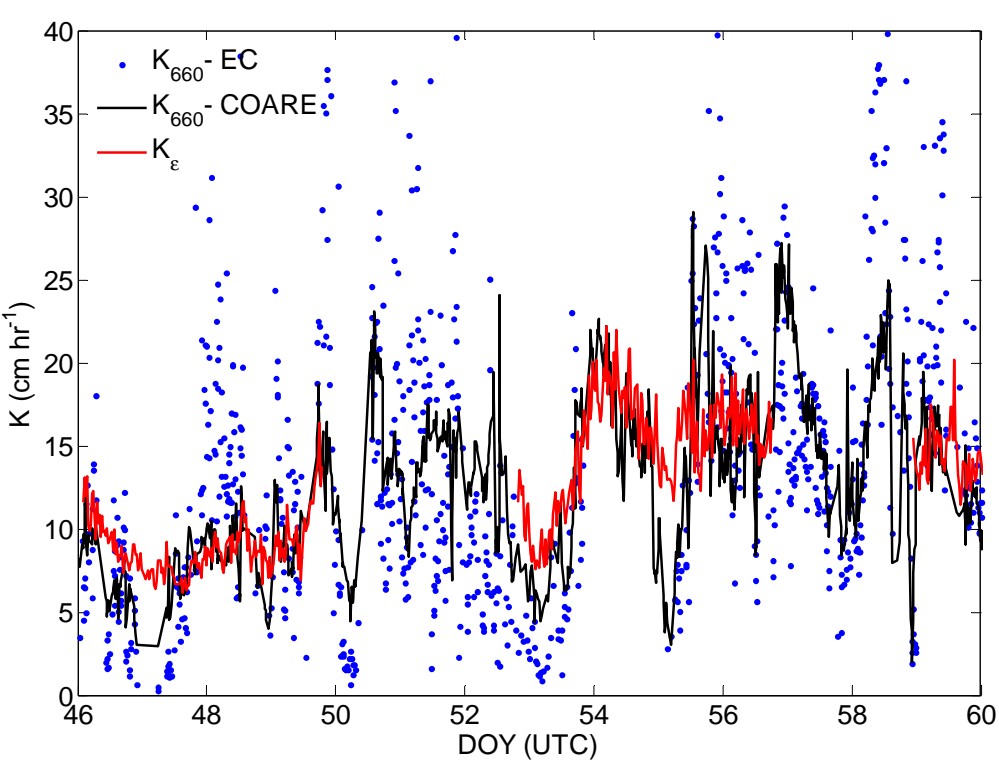

**Figure 9.** Gas transfer velocity, K, calculated from turbulent eddy dissipation rate ($\varepsilon$), COARE model, and EC 10-min data.