# Peer review of "Gradient flux measurements of sea-air DMS transfer during the Surface Ocean Aerosol Production (SOAP) experiment"

_Atmospheric Chemistry and Physics, 2017_

## Referee Comment (RC1) · D.T. Ho (Referee) · 11 Jan 2018

Smith et al. have written a nice paper with some interesting new results. They use 6 deployments of a catamaran during the Surface Ocean Aerosol Production (SOAP) study on the Chatham Rise to make atmospheric profile measurements of DMS at 4 different heights. These DMS measurements were then used with the gradient flux (GF) technique to determine DMS fluxes, and the results compared with previously published DMS fluxes determined using eddy covariance (EC) on the same cruise.

It is great to be able to compare independent measurements of gas fluxes (and gas transfer velocities) from the same cruise. Over the ocean, GF is a technique that has

not been used as frequently as EC due to some technical challenges, and Smith et al. seemed to have been careful with their data analysis, and to point out the caveats.

There are three significant results from this paper: 1) When u* from EC is used in the GF calculations (as opposed to bulk formulae u*), the DMS fluxes from EC and GF are comparable, 2) there are conditions under which the EC and GF data agree, but deviate from the NOAA COAREG parameterization, and 3) we are not at the point yet where we can parameterize gas transfer velocity with turbulent kinetic energy dissipation rate in the ocean.

The paper is laid out in a logical way, and written in a concise manner. I have no changes to suggest (except maybe make Figure 6 square, since it is a one-to-one comparison). I recommend that it be published as is.

---

## Referee Comment (RC2) · Anonymous Referee #2 · 31 Jan 2018

It is a pleasure to read such a clear and well-written manuscript. The authors present a detailed comparison between eddy covariance (EC) and gradient flux (GF) measurements of DMS flux from the southern ocean during the SOAP cruise. The introduction is very thorough and can bring most readers up to speed even if they are not familiar with some or all of the experimental and theoretical details. The experiments, analysis, and discussion are all very clearly presented, and the conclusions appear to be sound. With a paper like this, the reviewer's task becomes not "is this paper any good?" but rather "What (if anything) can I suggest to improve the manuscript?"

The one thing I expected to see in this paper is some discussion of the role that surfac-

tants might play in air-sea gas exchange. There were one or two places where surface buildup, microlayer enrichment (P. 15, l. 9) or near-surface gradients (P. 3, l. 13; P. 4, l. 32) were mentioned, but these seemed to refer more to the buildup of DMS itself rather than surfactants. Certainly in phytoplankton blooms in the nutrient-rich southern ocean there exists the possibility of large multi-functional organic molecules accumulating at the surface and influencing exchange across the surface microlayer. See Pereira et al., Biogeosciences, 13, 3981, 2016 for a recent reference; many earlier publications (quite a few of them referenced in that one) have also suggested that surface layer composition can have a significant impact on air-sea flux of trace gases. This should affect the EC and GF results roughly equally, and not significantly change the conclusions of the intercomparison between methods. But it will not be accounted for in the COAREG parameterizations, unless the surface layer were thick enough to affect the measured turbulent energy dissipation rate. (Which seems unlikely, and epsilon only appears in Eq. 4 to the $\frac{1}{4}$ power anyway.) In contrast, the discussion of the role of air bubbles is informative and seems complete.

During periods of atmospheric stability, the agreement between the two experimental methods but disagreement with the COAREG parameterization, is intriguing. Is it worth extending (maybe not in this paper) this analysis to include previous EC or GF gas flux measurements during periods of atmospheric stability? (Is there something fundamentally different about stable conditions, or is it just particular to these results?) Figures 5d and 7 show the discrepancy very clearly for this cruise.

Some more detailed and technical comments:

Abstract, l. 14 Is east of New Zealand really the southwest Pacific? I had always considered 180 E or W to be the central Pacific, but that is in the tropics. Not important at all; if the study location is generally considered to be the southwest Pacific then don't change it.

l. 16 might want to define "API" = atmospheric pressure ionization

[Figure]

P. 3, l. 34 hyphenate Monin-Obukhov

P. 4, l. 9, Is "closeby" a word? (Two countries divided by a common language?)

l. 14, Might want to mention that Liss and Merlivat is one of the earliest parameterizations of K, or something like that. Or maybe everyone knows that already.

l. 14 I really don't think "trialled" is a word. How about "performed trials" or "tried"? (See also P. 16, l. 31)

l. 21 "gas transfer parameterizations"?

P. 5, l. 17 "crowsnest" or "crow's nest"?

l. 21 "ship's"

l. 22 "Ecotriplet b660 backscatter" is a bit of jargon. Simplify ("backscattered light"), expand (to say what it's for), or delete?

P. 6, l. 7 "LES" is defined twice.

l. 22 and 25, Since the integrated stability function depends on z, don't you need to find the slope of the concentration profile vs. (ln(z) - Psi(z/L)) in order to calculate C*/kappa? (And Figure 4 shows that you did that too.)

P. 7, l. 23-26, Was the same aqueous DMS concentration used for both EC and GF analyses of K? (Which is perfectly reasonable, but means that intercomparisons of K between the two methods depend only on measured fluxes. The discussion of fetch effects in 4.3 is fine, no need to add much or anything here.)

P. 9, l. 14. I'm not sure how much validity there is in $R^2$ for only four data points. (Just an opinion; no changes needed.)

P. 10, l. 10-12. Would a plot of u* vs. u* for the two methods be useful? Probably just the summary here (along with Figure 6) is sufficient; I suppose the authors have already done this.

P. 11, l. 10, stratification leads to high fluxes? Walker et al., 2016 mention surfactants as possibly contributing to suppressed ventilation until winds pick up. But fluxes would be low until this occurs.

P. 12, l. 19 Could real-time motion correction be made for REA? (This doesn't need to go in the paper; just curious.)

P. 13, l. 3, "data are sparse"

P. 14, l. 29 "suppresses"

P. 15, l. 4, "near-surface"

l. 9, Could this be a place where surfactants are playing a role? (K from EC and GF lower than from using COAREG.)

l. 14, need to define K660

l. 18 "so they could have [had] a"

l. 19-21 True, there could be a bias. But if there were enhanced DMS at the ocean surface, it would lead to higher flux, and higher calculated K if using the bulk [DMS] at 5-6 m depth.

P. 16, l. 27, "The GF experiment/sampling equipment/whatever was deployed..."

P. 17, l. 18 First grant # looks like it is missing some digits.

Figure 1c. What is height above surface (or water level) on the spar buoy after it is deployed?

Figure 4 As mentioned earlier, the x-axis label text has the stability adjusted height. Might want to put something about the error bars in the caption (it is already in the text), because they are different between top and bottom even though they are both for DMS concentration or mixing ratio. Also would be good to make the tick labels, axis labels, and subplot titles a little larger. They are at the limit of readability now.

Figure 6 "Gradient Flux"? (y-axis label)

Figure 7 – put the arrow and DOY53 inside the axes or somewhere else. Right now they interfere with the x-axis label.

Figure 9 – also plot K-EC vs. K-COARE? And/or K-EC vs. Ke? This may or may not be informative. Figure 9 is pretty clear as is. And there is no need to make this paper any longer.
* * *

---

## Author Comment (AC1) · 28 Mar 2018

Response to Referee #1

RC : Smith et al. have written a nice paper with some interesting new results...

The paper is laid out in a logical way, and written in a concise manner. I have no changes to suggest (except maybe make Figure 6 square, since it is a one-to-one comparison). I recommend that it be published as is.

AC : We thank the reviewer for these positive comments. We have changed Figure 6 to have a square aspect ratio.

[Figure]

Response to Referee #2 RC : It is a pleasure to read such a clear and well-written manuscript. The authors present a detailed comparison between eddy covariance (EC) and gradient flux (GF) measurements of DMS flux from the southern ocean during the SOAP cruise. The introduction is very thorough and can bring most readers up to speed even if they are not familiar with some or all of the experimental and theoretical details. The experiments, analysis, and discussion are all very clearly presented, and the conclusions appear to be sound. With a paper like this, the reviewer's task becomes not "is this paper any good?" but rather "What (if anything) can I suggest to improve the manuscript?"

AC : Thankyou for the comments and careful reading of the paper.

RC : The one thing I expected to see in this paper is some discussion of the role that surfactants might play in air-sea gas exchange. There were one or two places where surface buildup, microlayer enrichment (P. 15, l. 9) or near-surface gradients (P. 3, l. 13; P. 4, l. 32) were mentioned, but these seemed to refer more to the buildup of DMS itself rather than surfactants. Certainly in phytoplankton blooms in the nutrient-rich southern ocean there exists the possibility of large multi-functional organic molecules accumulating at the surface and influencing exchange across the surface microlayer. See Pereira et al., Biogeosciences, 13, 3981, 2016 for a recent reference; many earlier publications (quite a few of them referenced in that one) have also suggested that surface layer composition can have a significant impact on air-sea flux of trace gases. This should affect the EC and GF results roughly equally, and not significantly change the conclusions of the intercomparison between methods. But it will not be accounted for in the COAREG parameterizations, unless the surface layer were thick enough to affect the measured turbulent energy dissipation rate. (Which seems unlikely, and epsilon only appears in Eq. 4 to the 1/4 power anyway.) In contrast, the discussion of the role of air bubbles is informative and seems complete.

AC : This is a good point, and while surfactants were not measured during SOAP, discussion of their potential effect has been added in the following places: Introduction

(p2, l32): "A further effect on gas exchange in winds under ∼10 m s-1 may arise from surfactants in the sea surface microlayer. Significant reductions in K have been measured in laboratory studies (e.g. Frew et al., 1990) and in open ocean measurements (Salter et al., 2011)."

Section 4.4 para1: "It is possible that these conditions of reduced mixing and convergence of biogenic material were conducive to an increase in surfactant activity in the sea surface microlayer, leading in to a reduction in K (Salter et al., 2011). This reduction would affect both EC and GF, but not COAREG."

Section 4.5 para 2: "Also the extrapolation will not allow for microlayer-based processes such as surfactants, which is consistent with the fact that on DOY 53 Ke did not decrease to the same extent as K derived from micrometeorological measurements."

RC : During periods of atmospheric stability, the agreement between the two experimental methods but disagreement with the COAREG parameterization, is intriguing. Is it worth extending (maybe not in this paper) this analysis to include previous EC or GF gas flux measurements during periods of atmospheric stability? (Is there something fundamentally different about stable conditions, or is it just particular to these results?) Figures 5d and 7 show the discrepancy very clearly for this cruise.

AC : The atmospheric stability is accounted for in the stability functions for GF, and these provide good agreement with EC. However, unlike a terrestrial situation, in this experiment the atmospheric stability was accompanied by surface-ocean stability which affects both the dynamics of aqueous transfer and the biogeochemistry. We certainly encourage a closer examination of previous gas flux measurements for similar effects, but as the referee suggests, this is beyond the scope of this paper.

RC : Some more detailed and technical comments: Abstract, l. 14 Is east of New Zealand really the southwest Pacific? I had always considered 180 E or W to be the central Pacific, but that is in the tropics. Not important at all; if the study location is generally considered to be the southwest Pacific then don't change it.

[Figure]

This term is in general usage, and in particular, is used in the published SOAP overview paper (Law et al., 2017), so it has not been changed here.

l. 16 might want to define "API" = atmospheric pressure ionization

The abbreviation has been defined: "atmospheric pressure chemical ionization mass spectrometry (API-CIMS)"

P. 3, l. 34 hyphenate Monin-Obukhov. Change made

P. 4, l. 9, Is "closeby" a word? (Two countries divided by a common language?). Corrected to "close by"

l. 14, Might want to mention that Liss and Merlivat is one of the earliest parameterizations of K, or something like that. Or maybe everyone knows that already.

This has been elaborated as : "one of the earliest parameterizations, Liss and Merlivat (1986)."

l. 14 I really don't think "trialled" is a word. How about "performed trials" or "tried"? (See also P. 16, l. 31) Changed to : "performed trials using"

l. 21 "gas transfer parameterizations"? "transfer" inserted

P. 5, l. 17 "crowsnest" or "crow's nest"? Corrected to "crow's nest"

l. 21 "ship's" Correction made

l. 22 "Ecotriplet b660 backscatter" is a bit of jargon. Simplify ("backscattered light"), expand (to say what it's for), or delete?

Expanded by the phrase "(an indicator of coccolithophores)", since it is of biogeochemical import for DMS.

P. 6, l. 7 "LES" is defined twice. Second definition has been removed

l. 22 and 25, Since the integrated stability function depends on z, don't you need to

find the slope of the concentration profile vs. (ln(z) - Psi(z/L)) in order to calculate C*/kappa? (And Figure 4 shows that you did that too.)

ln(z) has been explicitly expanded to: "(ln(z) -$\Psi c$ (z/L))"

P. 7, l. 23-26, Was the same aqueous DMS concentration used for both EC and GF analyses of K? (Which is perfectly reasonable, but means that intercomparisons of K between the two methods depend only on measured fluxes. The discussion of fetch effects in 4.3 is fine, no need to add much or anything here.)

This has been clarified in Section 3.3 by the addition of this text: "For both EC and GF, Cw was obtained from the same miniCIMS measurements, so that intercomparisons of K depend only on measured fluxes."

P. 9, l. 14. I'm not sure how much validity there is in R2 for only four data points. (Just an opinion; no changes needed.) P. 10, l. 10-12. Would a plot of u* vs. u* for the two methods be useful? Probably just the summary here (along with Figure 6) is sufficient; I suppose the authors have already done this.

The plot of u* from the two methods has been done, but in the interests of space has not been included. To indicate this, the following sentence has been added: "This slope is very close to that from a direct regression between u* from bulk fluxes and u* from EC."

P. 11, l. 10, stratification leads to high fluxes? Walker et al., 2016 mention surfactants as possibly contributing to suppressed ventilation until winds pick up. But fluxes would be low until this occurs.

The description of the stratification reported in Walker et al. 2016 has been expanded to include surfactants, to read: "Anomalously high DMS fluxes followed this, and Walker et al. (2016) suggest that the stratification provided optimal conditions for the accumulation of DMS in near-surface waters, through concentration of phytoplankton and reduced diffusive loss to sub-surface water, as well as possibly surfactant suppression

of ventilation. This accumulation would then have been ventilated to the atmosphere when winds subsequently increased and stratification was eroded."

P. 12, l. 19 Could real-time motion correction be made for REA? (This doesn't need to go in the paper; just curious.)

Conceptually it could, but the errors in estimating the real-time vertical motion of the platform are much larger than when postprocessing is an option.

P. 13, l. 3, "data are sparse" "is" changed to "are"

P. 14, l. 29 "suppresses" "supresses" changed to "suppresses"

P. 15, l. 4, "near-surface" "near surface" changed to "near-surface"

l. 9, Could this be a place where surfactants are playing a role? (K from EC and GF lower than from using COAREG.)

Yes, the following text has been added: "It is possible that these conditions of reduced mixing and convergence of biogenic material were conducive to an increase in surfactant activity in the sea surface microlayer, leading in to a reduction in K (Salter et al., 2011). This reduction would affect both EC and GF, but not COAREG."

l. 14, need to define K660

K660 has been changed to K, since all Ks presented have been normalised to a Schmidt number of 660. A statement to this effect has been added to Section 3.3: "K was also normalised to normalized to a Schmidt number of 660 ($CO_2$ at 25 oC) to facilitate comparison with other experimental results."

l. 18 "so they could have [had] a" Changed to "so they could have had a significant effect.."

l. 19-21 True, there could be a bias. But if there were enhanced DMS at the ocean surface, it would lead to higher flux, and higher calculated K if using the bulk [DMS] at

5-6 m depth.

Agreed. These two speculative sentences have been removed.

P. 16, l. 27, "The GF experiment/sampling equipment/whatever was deployed. . ." Changed to: "The GF sampling equipment was deployed. . ."

P. 17, l. 18 First grant # looks like it is missing some digits. Grant number corrected to: "grant numbers: 0851068, . . ."

Figure 1c. What is height above surface (or water level) on the spar buoy after it is deployed?

When deployed, the water level is approximately 1.6 m below the top of the yellow structure. A sentence has been added to the caption: "The sea surface is approximately 5.3 m above the base when deployed."

Figure 4 As mentioned earlier, the x-axis label text has the stability adjusted height. Might want to put something about the error bars in the caption (it is already in the text), because they are different between top and bottom even though they are both for DMS concentration or mixing ratio. Also would be good to make the tick labels, axis labels, and subplot titles a little larger. They are at the limit of readability now.

A sentence about the error bars has been added to the caption for upper and lower plots. "Error bars are the standard deviation of multiple samples." "Error bars are calculated from the residual error of the fit." The font size has been increased.

Figure 6 "Gradient Flux"? (y-axis label) Label on y-axis has been changed to "Gradient Flux"

Figure 7 – put the arrow and DOY53 inside the axes or somewhere else. Right now they interfere with the x-axis label.

This was an error during pdf conversion which will be fixed.

Figure 9 – also plot K-EC vs. K-COARE? And/or K-EC vs. Ke? This may or may not be informative. Figure 9 is pretty clear as is. And there is no need to make this paper any longer.

These were considered but we do not think these will significantly add understanding. And as the Reviewer states there is no need to make the paper longer. They have not been included.

References: Frew, N. M., Goldman, J. C., Dennett, M. R., and Johnson, A. S.: Impact of phytoplankton generated surfactants on air sea gas exchange, Journal of Geophysical Research: Oceans, 95, 3337-3352, doi:10.1029/JC095iC03p03337, 1990.

Salter, M. E., Upstill-Goddard, R. C., Nightingale, P. D., Archer, S. D., Blomquist, B., Ho, D. T., Huebert, B., Schlosser, P., and Yang, M.: Impact of an artificial surfactant release on air-sea gas fluxes during Deep Ocean Gas Exchange Experiment II, J Geophys Res-Oceans, 116, 2011.